# Effects of Multi-Level Eco-Labels on the Product Evaluation of Meat and Meat Alternatives—A Discrete Choice Experiment

**DOI:** 10.3390/foods12152941

**Published:** 2023-08-03

**Authors:** Anna Kolber, Oliver Meixner

**Affiliations:** Institute of Marketing & Innovation, Department of Economics and Social Sciences, University of Natural Resources and Life Sciences, Feistmantelstraße 4, A-1180 Vienna, Austria; a.kolber@hotmail.com

**Keywords:** multi-level labels, eco-labels, sustainability, willingness to pay, choice experiment, meat attachment, Hierarchical Bayes

## Abstract

Eco-labels are an instrument for enabling informed food choices and supporting a demand-sided change towards an urgently needed sustainable food system. Lately, novel eco-labels that depict a product’s environmental life cycle assessment on a multi-level scale are being tested across Europe’s retailers. This study elicits consumers’ preferences and willingness to pay (WTP) for a multi-level eco-label. A Discrete Choice Experiment was conducted; a representative sample (n = 536) for the Austrian population was targeted via an online survey. Individual partworth utilities were estimated by means of the Hierarchical Bayes. The results show higher WTP for a positively evaluated multi-level label, revealing consumers’ perceived benefits of colorful multi-level labels over binary black-and-white designs. Even a negatively evaluated multi-level label was associated with a higher WTP compared to one with no label, pointing towards the limited effectiveness of eco-labels. Respondents’ preferences for eco-labels were independent from their subjective eco-label knowledge, health consciousness, and environmental concern. The attribute “protein source” was most important, and preference for an animal-based protein source (beef) was strongly correlated with consumers’ meat attachment, implying that a shift towards more sustainable protein sources is challenging, and sustainability labels have only a small impact on the meat product choice of average consumers.

## 1. Introduction

The environmental sustainability of the food system has crucial roles in stabilizing the Earth’s system [1], mitigating climate change [2], and reaching the UN’s sustainable development goals [3]. The latest IPCC report stresses the high potential of demand-side actions to foster sustainable healthy diets that will contribute to “nutrition, health, biodiversity and other environmental benefits” [4]. A transformation towards more sustainable food consumption patterns can be supported by various instruments, including information provision, pricing, accessibility, and regulation of the food environment [5,6]. Providing information by label schemes stands out as a low-cost, easy-to-implement, and non-intrusive policy measure and enables consumers to identify the sustainability of products to support purchase decisions [7] as well as to encourage companies to improve their environmental standards [8].

In this study, the term “sustainability label” is used as an umbrella term and refers to four dimensions: environmental friendliness (such as organic or carbon footprint labels), ethics (such as animal welfare labels), social aspects (such as Fairtrade labels), and health aspects (such as nutrient-depicting labels) [9,10]. This study focuses on environmental aspects and defines the term “eco-label” as “a sign or logo that is intended to indicate an environmentally preferable product (…) based on defined standards or criteria” [11]. Currently, the Ecolabel Index [11] has a total of 456 eco-labels registered in 199 countries. Whereas binary labels guarantee a certain standard or not (label or no label), multi-level labels bring advantages, as such designs display intermediate qualities and hence provide more differentiated information in a simplified manner [12,13]. According to online consumer information platforms such as Standardsmaps.org or the German language platform Bewusstkaufen.at [14,15], the most prevalent sustainability labels on the European food market are binary labels. Examples are EU organic, Marine Steward Ship (MSC), Rainforest Alliance, Fairtrade, and the Carbon Trust Label [16]. Prevalent eco-labels in the empirical field of this study, the Austrian food market, are, for instance, Climate Partner, Carbon Trust, and various organic labels such as the EU organic label, the German organic label, and private associations (e.g., Demeter, Bioland) and manufacturers’ brands (such as the Austrian private organic label “Zurück zum Ursprung”, i.e., “Back to the Origin”, which identifies the sustainability performance of producers based on a sustainability assessment conducted by the Research Institute of Organic Agriculture FiBL [17]). Metric-labels or claims depict absolute values such as CO_2_-equivalents (in kg), challenging consumers to interpret the numbers. Only a few examples are available on the market [18].

The EU has now a clear focus on sustainability claims and labels. As part of the EU Green Deal, the European Commission announced the Farm-to-Fork-strategy (F2F) in 2019 and is currently working on guidelines for establishing a fair and sustainable food system [19]. Two ongoing workstreams, the green claims directive [20] and the sustainable food labeling framework [19], aim to ensure transparent communication on environmental claims across the EU and to harmonize how sustainability information related to food products is provided for consumers [19]. Already in use is a standardized approach developed by the European Commission for conducting life cycle assessments, the so-called product environmental footprint (PEF). In total, 16 criteria related to the environmental performance of a good or service are included in the calculations of the PEF [21].

As a reaction to the EU’s endeavors, European countries have started to develop and test eco-labels on the food market. Examples are the Enviroscore, the Eco-Impact, the Eco-Score, and the Planet Score [22]. These eco-labeling initiatives are based on the PEF but differ in terms of their calculation methodologies. They all have in common that the data are normalized, weighed, and then aggregated into a single score. The score is then translated into a multi-level design with an ABCDE scheme. This design resembles the earlier-developed multi-traffic label on food’s nutritional benefits, the so-called “Nutri-Score” [23]. The French Planet Score is especially interesting for this study, because its design is extended by the score of three subcategories, biodiversity, climate, and pesticides, which are issues that the French population is particularly concerned with, according to a representative survey [24]. Therefore, the label provides comprehensive information on a product’s environmental impact. It was developed because the French Eco-Score approach was not precise enough. Thus, the Planet Score tries to enhance methodologies aiming to include the environmental benefits of organic farming production methods. It was founded by the French Organic Food and Farming Institute and two research organizations, Very Good Future and Sayari. Currently, the Planet Score is being tested in selected French retail outlets and in Spanish Eroski stores. Further eco-label initiatives (such as Enviroscore, etc.) are currently being tested in retailers all over Europe at Lidl, Colruyt, Migros Switzerland, Coop Switzerland, Coop Sweden, and Carrefour [22].

Whereas research has focused on consumer preferences for binary labels in the past, few insights exist on consumers perceptions of multi-level labels [16]. Since providing information on food products depends on consumers’ reactions and preferences [5,25], this paper aims to investigate consumers’ preferences for multi-level labels on food products in Austria. This study examines how traffic light eco-labels (using the example of the Planet Score label) impact consumers’ perceived utility of and willingness to pay (WTP) for products with environmental benefits compared to binary labels (using the example of the Carbon Trust Label) (Section 1.2).

### 1.1. Literature Review

Research on visual sustainability labeling has focused on the effects that binary labels have on consumers’ psychological dimension. The results show higher WTP for food labeled, amongst others, with USDA organic, EU organic, animal welfare, Fairtrade, lower carbon footprint label, or fictional sustainability labels [26,27,28,29]. Studies show the positive utility of binary labels for consumers [30,31,32]. Grunert et al. [33] investigated consumer preferences in six European countries for multiple product categories including coffee, chocolate and ready meals and found that products that are labeled with Fairtrade, animal-welfare-approved, Rainforest Alliance, and carbon footprint labels have higher utility than non-labeled products. Sustainability labels can lead to greater product acceptance, as has been the case for chocolate associated with the Rainforest Alliance and the Brazilian Organic Seal [34]. Also, sustainability labels change the relevance of the price in both directions. Price is less important than organic and animal welfare label attributes for beef products in Germany and the US [26,29]. However, price is often a significant constraint to the effectiveness of labels. The importance of the price attribute has been perceived to be higher than that of eco-labels in numerous studies [30,31,35,36,37]. High prices are especially restrictive to repeat purchases of organic food items, as retail panel data revealed [38]. Additionally, sustainability labels increase the perceived healthiness and environmental friendliness of the product. This effect was found by Lazzarini et al. [39] in a study of nutrition claims, the country of origin, and organic labels on different protein sources, including chicken breasts.

Consumer research on multi-level labels is quite new; yet, there is a tendency for them to contribute to more sustainable food choices. The use of colors plays an important role in the effectiveness of eco-labels, according to Thøgersen and Nielsen [40]. When using traffic light colors for a carbon footprint design compared to black-and-white, the label’s effect on whether respondents chose the more sustainable coffee was intensified [40]. Products marked with green-colored eco-scores led to higher utility and more sustainable choices [41,42,43]. Red colored eco-labels decreased purchase intentions and prevented environmentally harmful choices among tested products including pizza margherita [42], meat balls, and lasagna [44]. Red, as a warning color, showed a stronger effect than green [42]. Label preferences resulted in higher WTP [45,46]. For instance, in a study by Sonntag et al. [9], out of several tested sustainability labels (Nutri-Score, animal welfare, organic) participants showed the highest WTP for whole milk labeled as having a low climate impact.

A successful impact of eco-labels would be the prevention of the consumption of foods that are especially harmful to the environment. For instance, global consumption of animal-based food has major impacts on the Earth’s system and climate change. The livestock sector accounts for 14.5% of all anthropogenic emissions [47]. According to previous studies, animal-based proteins emit considerably higher GHG emissions than plant-based meat alternatives: poultry (43%), pig (63%), beef from dairy herds (87%), and beef from beef herds (93%) [48]. Scholars and policymakers therefore advocate for a reduction in meat consumption [1,6], especially in Western countries [49] where meat consumption is deeply rooted in society [50]. Eco-labels can draw attention to more sustainable “meat alternatives”, products that try to imitate animal-based products in all sensory aspects based on environmentally friendly proteins [51,52]. Shifting from niche to mainstream, the meat alternatives market in Europe is predicted to grow from 1.5 € bn in 2018 to € 2.4 bn by 2025 [53].

Hybrid meats are a compromise, as they reduce meat consumption by adding vegetables to the product [43]. Since meat reduction plays an essential role in adapting to a sustainable food system, the present study tests how consumers react to eco-labels depicted on minced meat products with different protein sources: meat-based, plant-based, and hybrid (meat and vegetables).

### 1.2. Hypotheses Development

The effectiveness of sustainability labels depends on multiple factors ranging from individual factors including altruism [54], environmental attitudes, environmental concern (EC) [55], sociodemographic factors (gender, age), etc., to label characteristics, to context factors such as the product type, origin, and price [16]. Another individual factor is consumer understanding of the presented information [56,57]. Whereas general environmental knowledge was, in some studies, found to be relevant for predicting green consumer behavior [58], context-specific knowledge on the environmental performance of products and labels seems to be a fundamental requirement that allows reasoned and well-informed choices [59]. Taufique et al. [57] support the importance of specific knowledge and found that perceived eco-label knowledge (ELK) has an indirect positive effect on pro-environmental consumer behavior. Also, in the study by Grunert et al. [33] the label effectiveness of the Fairtrade and Carbon Footprint labels was shown to depend on consumers’ understanding. The objective of testing the Planet Score’s effectiveness (positive vs. negative evaluation) led to hypotheses H1a (preference) and H1b (importance).

**H1a:** 
*Planet Score B (vs. Planet Score D) is more preferred by respondents who perceive themselves to have higher eco-label knowledge.*


**H1b:** 
*Higher eco-label knowledge positively influences the importance of eco-labels.*


Furthermore, label effects can be explained by consumers’ attitudes towards sustainability issues. Ghvanidze et al. [60] showed that consumers’ attitudes are in line with their preferences, as highly environmentally conscious people, in particular, value ecologically and socially responsible produced food. Thøgersen and Nielsen [40] found that consumers with high EC are more prone to choosing the “responsible” product (in their study, this was coffee with a low carbon footprint). The more respondents were concerned about the environment, the higher the probability of choosing coffee labeled with a green (vs. red) colored footprint. Similarly, the authors supposed that EC positively influences the preference for a positively evaluated Planet Score [40]. Therefore, we developed H2a to see if more environmentally conscious respondents prefer Planet Score B (environmental impact is rather low) over Planet Score D (environmental impact is rather high) and H2b to see if the importance of eco-labels is also dependent on EC.

**H2a:** 
*The Planet Score B (vs. Planet Score D) is more preferred from respondents who are more concerned about the environment.*


**H2b:** 
*Higher environmental concern positively influences the importance of eco-labels.*


The health of the environment is interconnected with the health of human beings [4]. Research has found that the more respondents are concerned with health, the more they choose products with environmental benefits. For instance, organic food consumers are relatively more concerned about their health than consumers buying conventional food [61]. Health-conscious respondents are prone to choosing products with sustainability labels such as sustainable palm oil (RSPO) [62], palm-oil free [63], as well as health and nutrition claims [60,64]. Therefore, H3a investigates if health consciousness (HC) affects the preference for Planet Score B vs. Planet Score D and, in accordance with the above considerations, H3b investigates the importance of HC.

**H3a:** 
*Planet Score B (vs. Planet Score D) is preferred by respondents who are more concerned about their health.*


**H3b:** 
*Higher health consciousness positively influences the importance of eco-labels.*


The protein source of meat products (e.g., beef, pork vs. plant-based meat alternatives) [65] has been shown to be relevant for food choices in discrete choice experiments, next to other factors such as price [29,60], national or local origin [36,65], and quality labeling (i.e., USDA) [29]. The protein source is relevant to this study, because plant-based or hybrid meat could contribute to a transition towards a more sustainable food system. Whereas food neophobia and familiarity have stronger impacts on the acceptance of novel products such as cultured meat or insect-based products [66,67], meat attachment—describing a respondent’s emotional bond towards meat consumption—seems to be more relevant concerning the adoption of plant-based meat alternatives [68,69]. Meat consumption is deeply rooted in European society [70], leading to H4.

**H4:** 
*The higher the meat attachment of respondents is, the lower their preference for meat alternatives will be.*


The objective of this study is to test consumer preferences and WTP for specific sustainability labels. Because there is no multi-level eco-label on the Austrian food market available, potential effects of such a label were tested on the Austrian population.

## 2. Materials and Methods

An online survey was conducted in March 2023, in which data were collected through a professional online panel provider that allows the anonymous recruitment of participants according to preselected criteria. The Austrian population was represented by applying a selection filter with the quota parameters “age”, “gender”, and “education”. Before launching the survey, a pretest was conducted to test the internal validity of the empirical design (n = 50). In the final survey, a total of 632 respondents participated, 23 had to be excluded for incomplete responses, and 73 were excluded for failing the attention check, leaving a final sample of n = 536 (response rate = 84.8%). For the discrete choice experiment (DCE), respondents who only chose the no-choice option (28) and no-minced-meat consumers (animal- or plant-based) still remaining in the sample (18) were excluded to guarantee that the answers reflected respondents’ own preferences. Consequently, a sample of 490 respondents was available for the analysis of the DCE. Table 1 provides an overview of the participants’ sociodemographic data in comparison to the Austrian population. The sample structure is very similar to the structure of the Austrian population, even the proportion of meat eaters vs. vegetarians/vegans is close to the overall distribution. The quotas of the relevant sociodemographic variables (gender, age, place of living, education, and household income) match between the sample and the statistical population. Thus, we are convinced that results are transferable to the overall Austrian population.

The online survey had the following structure: introduction, warm-up questions, information on eco-labels, DCE choice sets, explanatory scales, and sociodemographic questions. After an introduction page including a data protection notice, participants were asked about their food consumption of meat and meat alternatives. One control question was asked on minced meat consumption: “How often have you eaten dishes with minced meat (animal or plant-based) in the last year?” with the answers: at least once a week, several times a month, once a month, less than once a month, never. As mentioned before, respondents who stated “never” were excluded from the DCE. An information part followed to shortly explain the three labels, which were in accordance with real-life labels but were self-designed: the Eco-Score label (comparable with the designs of the French Planet Score) and the Climate Protection label (which is close to the Carbon Trust label). Although not being part of the DCE, a further existing eco-label was included in the explanation part (the Austrian “PrüfNach” label) in order to prevent an attention bias. The DCE was introduced with the explanation of a hypothetical shopping situation with the following wording: “Imagine you are grocery shopping and standing in front of the refrigerated counter. You want to buy minced meat and see available products. We ask you to choose your preferred product in each of multiple rounds. If you normally do not buy minced meat for yourself, imagine choosing for someone else. It is also possible to make no choice.” The instructions were adopted from comparable studies, including Apostolidis and McLeay [64] and Sonntag et al. [9]. Based on the product attributes, a reduced study design was calculated by means of the Microsoft Excel add-in XLSTAT (Version 2018.1.1). Based on the predefined product attributes and attribute levels, XLSTAT produced an optimized and balanced output of 12 choice sets. Each attribute level of each product category had the same frequency (see Appendix A
Table A1). Respondents passed the 12 choice sets (example Figure 1; Appendix A
Table A2), each presenting three items and a no-choice option, allowing choices to be closer to true preferences [76]. In addition, a 13th choice set, a holdout choice set, was included in the study design, which was not used to approximate partworth utilities, but to see if the choice in the 13th set could be replicated by using the approximated partworth utilities (i.e., the “hit rate” [77]). Based on the maximum utility choice rule [77], the choice of each respondent between the alternatives of the 13th choice set and the no-choice option can be predicted. If the hit rate is much lower than one (and close to the random probability of 0.25), the test design is invalid. Moore [77] identified a maximum hit rate of around 0.7 from the literature, and this was used as a threshold for our study.

Subsequently, the participants’ knowledge and motives (see Appendix A
Table A3) were surveyed using a seven-point response scale ranging from 1 = “I totally disagree” to 7 = “I totally agree”. Respondents answered four items on subjective ELK [57,78,79] and five items on environmental concern (EC) [80,81] adapted from the New Ecological Paradigm Scale developed by Dunlap et al. [82]. To measure meat attachment, one to two item(s) of each factor (hedonism, affinity, entitlement, and dependence) were extracted from confirmatory analyses tested by Graca et al. and Kühn et al. [83,84], resulting in a seven-item scale. An attention check was integrated within the last scale on HC, a six-item scale based on the general health index scale from Roininen et al. [85]. The attention check consisted of one item providing the instruction, “This question is for checking your attention, please tick ‘strongly disagree’”. Therefore, respondents who were not reading thoroughly or answering randomly could be sorted out. Finally, the sociodemographic variables gender, age, residence, education, and household income were asked about. The questionnaire was in German language, and the mentioned constructs from the literature were translated and improved by feedback gathered from the pretest. Subsequent analyses were conducted in XLSTAT (Version 2018.1.1) and the software solution SPSS (Statistical Package for Social Sciences, version 26).

*Experimental design and Estimation of WTP:* A common approach for evaluating consumer preferences and WTP on food attributes is the application of a DCE [86]. Numerous studies have used the DCE with focuses on meat attributes [29,36], the type of protein [65,87], the country-of-origin [88], and label preferences [9,28]. The advantages of DCEs are that realistic buying situations are simulated, “where consumers choose between one or more products from a restricted product set (evoked set)” [89]. Respondents are supposed to choose the most beneficial product for them. They have to make a tradeoff between desirable and undesirable attributes which makes the results strongly related to the actual market share [90]. Furthermore, DCEs can provide results with high external validity as they reduce respondents’ hypothetical bias (i.e., deviation between stated and actual behaviors) [91].

The following attributes were included in the study: eco-labels, production condition, protein source, origin, and price (Table 2). Minced meat was chosen as the product category because unlike whole meat cuts, there are reasonable product depths of both meat-based and plant-based products available at Austrian retailers. The product category is well-known and accessible for all population groups [43,65,92]. To rebuild products with realistic prices according to the market conditions, a store check was conducted on the 21st and 22nd of December 2022 in Vienna. Therefore, the three main retailers, representing around 90% of the Austrian retail market share, were visited [93,94]. Price attributes and protein sources based on the product range from the store check were defined as between 3.59 € and 5.99 € per 400 g and included the protein sources beef, hybrid, and plant-based with pea protein [43,87]. Up until now, hybrid meat has been offered in Austria online only (beef and pea protein 50/50). This has gradually made meat reduction more accessible to consumers [65], and it is thus part of this study. Two eco-labels were included in the experimental design, the binary label “Carbon Trust” with the claim “CO_2_ Reduced” referring to the company’s measurement and reduction of the product’s carbon footprint [33,95,96]. For the study we used a self-designed Climate Protection label and referred to it as the “Carbon Trust label”. Second, as a comprehensive multi-level label, an adapted design of the French Planet Score label was used [97] which, in this study, is referred to as the “Planet Score”. The Eco-Score includes multi-traffic light (MTL) scores ranging from A–E: an overall score and ratings for the three subcategories climate, water protection, and biodiversity [1]. Also, these issues are more tangible to consumers compared to more complex topics such as eutrophication [98,99]. The Planet Score is either shown with a relatively good (B) or a relatively bad (D) overall rating, comparable to the study of Sonntag et al. [9]. To fulfill the DCE requirement of independent attributes [100], Planet Score grades were not linked to the actual product’s environmental impact, which is a slight deviation from the objective grading of food products. Beef, for instance, should not be graded with Score B, as GHG emissions are, in general, quite large for beef production [98]. Previous studies found that the environmental impacts of both animal and plant-based ingredients can vary widely and that there is a range where impacts can overlap [48]. Thus, it may be realistic that under certain production conditions (intensively farmed beef emits around three to four times less CO_2_ equivalents compared to an extensive production system [101]), beef might be graded as mediocre or relatively good. The ratings of the subcategories have small deviations in accordance with the overall rating found in the literature [99]. Further relevant product attributes for consumers are the production conditions, organic and conventional [36,65,95], and the origin of production, “Austria” and “within the EU” [31,37].

This study referred to the Random Utility Theory (RUT) which was first proposed by Thurstone [102] as a theory of paired comparisons (comparing pairs of choice alternatives) and was later extended by McFadden [103] to a theory of multiple comparisons. The RUT calls “utility” a latent construct, saying that the utility for each choice alternative exists in consumers’ heads but cannot be “seen” by researchers. More concretely, the total utility *U_in_* that an individual *n* associates with the alternative *i* is the sum of the systematic (observable) *V_in_* and random (unobservable) utilities (*ε_in_*), as shown in Formula (1).
(1)$$U_{in}=V_{in}+\varepsilon_{in}$$

The deterministic component was assumed to be linear *V_in_* = β∙*X_in_*, whereas *X_in_* is the vector of the observable product attributes, and β represents the mean preferences of the respondents for each attribute [45]. For this study, the utility *V_in_*, was assumed to be the linear function [65] of the protein type, eco-labels, origin, production method, and price. By integrating the selected product attributes, the following utility function of a consumer *n* for the alternative *i* was approximated according to the additive model shown in Formula (2).
(2)$$U_{in}={\beta 1}_{protein\;type}+{\beta 2}_{eco-label}+{\beta 3}_{production\;method}+{\beta 4}_{price}+\varepsilon_{in}$$

Based on the results of the respondents’ DCE choices, the partworth utilities of the attribute levels and the relative importance of each of the attributes were approximated. For hypothesis testing, the Hierarchical Bayes (HB) estimation is used to approximate individual partworth utilities [104,105]. Considered to be state-of-the-art in food research, the HB approach allows statistical efficiency in data processing [86]. Quality loss during the process of estimation, including the local optima and convergence problems, is avoided [106].

Rooted in welfare economics, the WTP is a concept that describes the marginal rate of the substitution of certain attributes for price levels [107], that is, how much consumers are willing to pay for a particular product attribute if all other attributes remain constant. With reference to the additive compensatory decision rule in Formula (2) [108], WTP can be expressed as the ratio between the negative utility per attribute level β*_attribute_* and the utility per money unit β*_price_* [108,109] (Formula (3)). Any change in *U_in_* due to a variation in the attribute levels can be substituted by adapting the price accordingly.
(3)$$WTP=-\frac{\beta_{attribute}}{\beta_{price}}$$

Using DCE for the approximation of *WTP* is a common approach in consumer research [108] and has been applied within a vast scientific body of comparable food studies [29,32,35,95,110].

The hypotheses were tested by applying the Pearson correlation between the explanatory variables and the results from the DCE. A more detailed description can be found in Section 3.3 Hypotheses Testing.

## 3. Results

### 3.1. Results of the DCE

The validity of the DCE is assumed to be very high based on the following approach: After approximating the attribute utilities (the utilities of each attribute sum up to 0), the prementioned 13th choices were replicated on the basis of the maximum utility theory. It was expected that the choice with the highest overall utility—including the no-choice option—would be selected. The hit rate amounted to 0.80 (n = 490), which is much higher than the threshold we defined, confirming the result of Moore [77] with 0.7. This means that 80% of all choices were predicted correctly. For all other choices (1 to 12), the hit rate was calculated as well; it was even higher (between 0.85 to 0.93) and amounted to 0.90 in total (5260 of 5880 choices from 490 respondents were predicted correctly). This result is not surprising, as the choices form the basis for the approximation of partworth utilities by means of HB. The most frequently chosen product card (profile 8, 1031 times; only regular choices 1–12 were analyzed here) was minced meat with beef from the EU, with organic production showing the lowest score (Appendix A
Table A4). The most frequently chosen profiles were all based on beef, which is a clear indication that the protein source is of high relevance and beef is the preferred choice, as we observed in the following analysis. The no-choice option was used frequently (1161 times), which might also be due to the fact that most of the respondents clearly preferred meat. Out of the choices, the partworth utilities *u_i_* were approximated by means of HB estimation, confirming Formula (2) (additive model). The results from the DCE included partworth utilities and the relative importance of the attributes (Table 3). The latter, the significance of the attribute level for a stimulus’ total utility, can be specified by means of the partworth utilities. A partworth utility itself does not indicate the relative importance of the attribute and whether an attribute would contribute to a change of preference or not [100]. It is the difference between the minimum and maximum utilities per attribute that matters: the higher the difference is, the more importance the relevant attribute will have.

Consumers highly valued the protein source beef (*u_i_* = 3.35), followed by the lowest price 3.59 € (*u_i_* = 1.80), and the eco-label “Planet Score B” (*u_i_* = 1.15). Positive partworth utilities were also approximated for the Austrian origin (*u_i_* = 0.62) and organic production (*u_i_* = 0.23). As expected, price had a negative influence on the simulated buying decision, as high prices are usually less attractive to consumers than low prices. The price function amounted to β_price_ = −1.57, which means that an increase in the price by one Euro reduces the utility of the alternative by −1.57. Regarding eco-labels, the highest *u_i_* was approximated for the label “Planet Score B” (*u_i_* = 1.15), followed by the label “climate protection—CO_2_ reduced” (*u_i_* = 0.22). Remarkably, a product labeled with the Planet Score D label (*u_i_* = −0.37)—signaling a negative environmental impact—had a higher partworth utility than a product without any sustainability certification (*u_i_* = −1.00). Compared with the beef attribute (*u_i_* = 3.35), the protein-source “plant-based (peas)” resulted in a particular low utility (*u_i_* = −2.88), whereas the hybrid beef and plant-based (50/50) product showed an intermediate negative average utility score (*u_i_* = −0.47).

The attribute with the highest relative importance was the protein type (44.2%), followed by price (24.7%) and eco-labels (17.7%). The importance of the protein source is undeniable when it comes to choosing between animal-based and alternative protein-based meats. Austrians obviously still prefer meat which makes the other product features much less important. Altogether, our results show the highest importance for the intrinsic attributes (protein source, price) and the lowest for the extrinsic attributes (origin, production condition, sustainability) of minced meat products.

### 3.2. Willingness to Pay (WTP)

The WTP was derived to show consumers’ readiness to pay an average premium of +0.73 € for a Planet Score B ($=-\frac{\beta_{Planet\;Score\;B}}{\beta_{price}}=-\frac{1.15}{-1.57}$) and +0.14 € for a Climate Protection—Reducing CO_2_ labeled product. A product labeled with a negative impact on the environment would need a discount of −0.23 €, which is less than the discount of −0.64 € required for non-labeled products. For an average consumer, plant-based pea protein as a protein source would need a discount of −1.84 €, whereas the WTP for its beef equivalent is +2.14 €. Table 4 summarizes the WTP for minced meat attributes.

The mean, however, does not tell the whole story. Obviously, the majority of consumers prefer beef. The average WTP of −2.14 € simply tells us that, on average, Austrian consumers are not willing to switch from meat to plant-based alternatives. They are still carnivores. But, there is a significant proportion of consumers who prefer plant-based alternatives, at least to some extent. The proportion of respondents with a positive partworth utility for beef sums up to around 80%, which more or less corresponds to the negative partworth for a plant-based meat substitute. Likewise, there is a proportion of about 20% with a negative partworth utility for beef and a positive partworth for plant-based meat substitutes (and, therefore, a positive WTP) (Table 5). If we only consider those respondents with *u_beef_* ≤ 0 (n = 107), the WTP is positive for plant-based +0.95 € and 50:50 (+0.21 €) meat alternatives. For both groups, those with a very high preference for meat (*u_meat_* ≥ 2; *u_plant-based_* ≤ −2) and those who prefer plant-based alternatives (*u_meat_* ≤ −2; *u_plant-based_* ≥ 2), the importance of the product attribute “protein type” is of the highest importance with an average importance rate beyond 50%. All differences between these groups are highly significant (*p* < 0.001). The effect size is, particularly for the attribute protein source, very high (*η*^2^ = 0.63 and 0.59, respectively; Appendix A
Table A5).

### 3.3. Hypotheses Testing

The reliability of the hypothetical constructs ELK, EC, MA and HC was tested by means of Cronbach’s alpha (CA). All items were kept on a relevant scale due to the excellent internal reliability [111] (Appendix A
Table A3). New parameters were created by calculating the mean values, excluding respondents who missed one or more items out. Table 6 contains the descriptive statistics for all explanatory variables, including the CA, mean, median, standard deviation (SD), min, max, and N.

To prove hypotheses H1–4, correlation analyses were conducted between the explanatory variables and the results from the DCE. First of all, we can see that the factors ELK, EC HC, and MA are inter-related (Table 7). Generally, Pearson’s correlation coefficient *r* was quite high. On one side, EC and HC were significantly positively correlated (*r* = 0.58 ***). Respondents with higher awareness of environmental issues also tended to have healthier lifestyles. The attachment to meat consumption (MA), on the other side, was significantly and negatively correlated with the other factors (*r* = −0.43 ***); respondents who were very attached to meat tended to be less cautious in their views of ecology and health. This is particularly relevant for our study, as we assumed that MA might have an influence on the preference for meat alternatives (H4).

Contrary to our hypothesis H1a, H2a, and H3a, the higher utility for the multi-level label Planet Score B compared to Planet Score D could not be explained by the tested constructs; *r* was insignificant for all constructs in Table 8. The respondents’ preference for the Planet Score B label was independent of their eco-label knowledge, environmental concern, and health consciousness. Thus, the hypotheses H1a, H2a, and H3a were rejected.

This further analysis clearly shows that preferences for animal-based protein sources are strongly influenced by MA. The personal emotional bond towards meat consumption drives preferences for beef and aversion to plant-based protein, leading to the acceptance of hypothesis 4: The more respondents are attachment to meat, the less they prefer meat alternatives (*r* = −0.58 ***) and the higher their preference for meat is (*r* = 0.51 ***). This relation was the strongest within all constructs. MA obviously has the largest impact on consumers’ decision to accept or reject meat alternatives. Figure 2 visualizes these relationships by a positive correlation between (a) MA and the utility of beef and a negative one between (b) MA and the utility of plant-based alternatives. There are some exceptions from these overall tendencies, in particular, on the upper left side in (a) and the lower left side in (b), but in general, the assumption holds.

As for the preferences for different protein sources, we can see that there are also influences of EC and HC on the preference for plant-based protein (*r* = 0.34 *** and 0.29 ***). In contrast, the preference for meat decreases with higher EC and HC values to almost the same extent (*r* = −0.37 *** and −0.29 ***; see Table 8).

Concerning H1b, H2b, and H3b, the perceived importance of the attribute eco-label was significantly correlated with the tested constructs on respondents’ motives and knowledge on ELK, EC, and HC. In particular, EC (H2b) seems to have a higher impact on the importance of eco-labels with *r* = 0.34 *** (Table 9). These hypotheses were therefore confirmed, even though Pearson’s correlation coefficient *r* was rather low. Eco-labels gain importance the higher the perceived ELK, EC, and HC are. The constructs are able to explain (at least to some extent) why eco-labels are preferred and are perceived as more important for some respondents compared to other beef attributes (production condition, origin; for the latter attributes, correlations are low and, in most cases, not significant). In addition, MA is obviously a significant construct that is able to explain why consumers are rejecting meat alternatives, as we showed above.

## 4. Discussion and Conclusions

Our results shed new light on consumer preferences and WTP for multi-level vs. binary eco-labels. To our knowledge, it is the first study with a multi-level label that comprises subdimensions—climate change, biodiversity, and water usage. Especially, when consumers lack time in shopping situations, comprehensive multi-level labels (such as the Planet Score) can reduce confusion and information overload by bundling information and presenting it in an easy and self-explaining manner [45]. The implementation of novel labels such as the Planet Score strongly depends on consumers’ preferences and reactions to the label [25]. In line with previous studies, our results reveal that eco-labels can effectively influence consumers’ choices, at least to some extent. In our study, an environmentally friendly product was preferred over an environmentally harmful one. This is consistent with what has been found in previous studies on comparable eco-labels [41,42,65]. The multi-level label Planet Score B was preferred over the binary Carbon Trust label, which indicates that colorful designs with scores rated from A to E bear certain advantages to consumers. This is supported by the study by Thøgersen and Nielsen [40], who showed that using traffic light colors for a multi-level carbon footprint design improved the label’s effect in comparison with a simple black-and-white footprint design. Choices for sustainable products were intensified. In line with the studies by Carlsson et al. [44], and Rizov and Marette [42], the respondents of our study tended to avoid red-light eco-labels. Contrary to Sonntag et al. [9], who argued that manufacturers and retailers cannot afford negative sustainability labels, our results show that even a negative Planet Score is preferred over no eco-label. This could be an incentive for manufacturers and retailers to enhance their environmental sustainability throughout the life cycle of food products and communicate their progress transparently.

Despite respondents’ preferences for eco-labels, our results point out that other product attributes such as the type of meat (beef, plant-based) and price are more decisive than labels on environmental sustainability. This is in accordance with the literature [33,65] and emphasizes the limitations of labeling as a policy measure. In view of the huge importance of the attribute level “meat”, the results from this study are supported by Apostolidis and McLeay [65], who examined consumer preferences for minced meat and found that the attribute protein source has the highest relative importance (as in our study), and “meat-free” is the least preferred attribute among other attribute levels of protein sources (i.e., beef, pork, etc.). For consumers who are expecting real meat if they buy minced meat products, plant-based alternatives are not a real alternative, which also explains the high proportion of no-choice answers in our data. Whenever only plant-based choices (or mixed meat) were offered (and/or the meat alternative did not correspond to the expectations of the respondents), the no-choice option was selected. However, we also identified an important target group within the sample that preferred plant-based alternatives. In contrast to the overall result of the study, plant-based alternatives are gaining a positive, significantly higher partworth utility compared to minced meat from beef (and, of course, the protein source is much more important for this group). We have to consider this result when interpreting the overall (negative) partworth utility of the plant-based alternatives.

Previous studies found that other attributes, such as nutritional information [33], fat content [65], and animal welfare [45], were more important than comparable environmental sustainability claims. We found that eco-labels were more highly valued than the country of origin—a clear contradiction to the literature [46,65]—and the production condition was more highly valued when it came to food choices, which is inconsistent with the results of Feucht and Zander [46] but in line with those of Sonntag et al. [9]. Despite consumers showing high awareness for organic production in Sonntag et al. [9], and potentially because of the fact that organic products are well-established and available in every supermarket, Janßen and Langen [112] argued that organic claims and labels cannot compete with the attractiveness of novel and colorful eco-labels. This may be one reason for eco-labels outperforming the country of origin, too.

The effectiveness of eco-labels is strongly related to consumers’ WTP a premium for eco-labeled foods, which are, in general, more expensive [25]. In our study, price had a significant impact on product choice with a negative price function implying a shrinking utility for higher prices. The highest WTP regarding the studied eco-labels was identified for a positive evaluated Planet Score (B) (+0.73 €), moderately high for Carbon Trust (+0.14 €), second lowest for a negatively evaluated Planet Score (D) (−0.23 €), and lowest for no label (−0.64 €). This result ties well with previous studies [7,9,45,46], wherein a high WTP for food products was associated with a similarly positive eco-label (climate label). Compared to a non-labeled product, consumers need less of a discount for an environmentally harmful labeled product (Planet Score D). This finding may be explained by a positive association with eco-labels (independent from their message) or by unfocused processing of the provided information. A similar conclusion was made by Janßen et al. [112] and Loo et al. [113]. It could be due to a lack of understanding of the respective label that respondents showed higher preference for both labels being depicted together (in this case, organic and GMO free or animal welfare), even though information from both labels might be redundant.

Red, as a warning color, can have a stronger effect than green-colored eco-labels on purchase intentions [42]. Our results show that the red-light Planet Score attribute decreases a product’s utility. Furthermore, the analysis found clear evidence of the perceived importance of the attribute eco-label correlating with respondents’ ELK, EC, and HC. Consequently, if the Planet Score is introduced in the Austrian market, it may become successful if it is targeted at consumers who are highly aware of the environmental and health issues associated with their food choices.

The results from our study provide relevant insights into consumer preferences and WTP for a multi-level eco-label (for the example of the Planet Score). The effectiveness of eco-labels faces certain challenges. On the consumption level, there are doubts about the real effectiveness of eco-labels and little evidence on changes in food behavior patterns [8,114]. Consumers’ lack of awareness of, and knowledge about, eco-labels is due to insufficient promotion [95,115] and consumers’ green skepticism (not trusting information on a product’s sustainability) [57,116,117]. These assumptions from the literature, such as the prioritization of other product attributes over eco-labels, such as price [16], are in line with our results.

Our study clearly reflects the importance of meat consumption in Western societies, as choosing beef as protein source was strongly correlated with respondents’ meat attachment. As expected from Graça et al. [83], the emotional bond towards meat consumption reduces the choice of plant-based alternatives and is a barrier to shifting towards more sustainable diets. Other reasons for the approximated low utility of plant-based alternatives may be lack of familiarity, food neophobia, or lower perceived quality [66,67]. In line with the study by Edenbrandt et al. [43], the utility of hybrid products containing beef and pea protein lies in between those of beef and plant-based alternatives and may be a compromise and means to overcome meat attachment (at least in the long run). Changing individual behavior towards eating less environmentally harmful protein sources appears to be challenging, as meat consumption is deeply rooted in Western society and is largely perceived as “nice, necessary, natural, and normal” [50]. Also, in our study, the majority of the respondents’ were classified as carnivores. These consumers will rarely change their diet patterns, at least in the short term.

The interpretation of our results has to consider several limitations. One concern about our study design is that the opt-out option obviously had a positive utility for respondents, revealing that the no-choice option was often preferred over the presented choice alternatives (in particular, if the 100% meat alternative was not part of the choice set). As a consequence, the attribute “protein source” had a very high level of importance. Including only meat in the experimental study design (e.g., beef, pork, mixed) could probably have a significant influence on the importance of the other attributes, which would gain importance.

Furthermore, the study design contained a few rather unrealistic options (such as organic and cheapest price or plant-based and Eco-Score D). This limitation was not a big issue, because although these combinations do not represent products that consumers would expect in their everyday shopping behavior, they are still possible. The results should, however, be interpreted carefully, as the restricted choice set of our DCE cannot be compared with a supermarket’s wide product range and multiple different attributes [5].

The Planet Scores were not based on life cycle assessment due to the aspired independency of the attribute groups. There is currently a lack of publicly available information on each product’s supply chain. Retail settings require dependable life cycle assessments and product ingredient data for implementation [41]. However, these inaccuracies would not affect this study’s key findings, which seem to be quite realistic. The importance of eco-labels is not predominant. For consumers, there are more relevant product features, such as the product source or price.

As the Planet Score is not available and the Carbon Trust is not widely available on the Austrian food market, and because the choice situation is not a real purchasing situation, a hypothetical bias, such as respondents overstating their WTP, may have appeared [118]. Also, an attention bias may have arisen from explaining the meaning of the two labels at the beginning of the study. Future studies may also consider a smaller number of presented choice sets (if possible, without reducing the robustness of the DCE) to better maintain respondents’ attention spans and reduce potential “fatigue effects”. It could also be wise to split the choice sets into two separate sets and put the choice sets at different positions in the questionnaire, in particular if a large number of choice sets is necessary. This approach might further reduce fatigue effects and increase the attention of respondents during the choice task. However, the latter is probably not always compatible with the structure of the questionnaire. Because this experiment focused on minced meat, future studies may explore the effects of a multi-level eco-label on other product categories such as staple food, convenience food, snacks, or beverages. It remains unclear how much the subdimensions of the Planet Score are relevant to consumers’ food choices. We identified an important target group within the sample preferring plant-based alternatives. Detecting clusters of consumers’ preferences, for instance, by Latent Class Models would overload this paper and leaves room for future projects. Future research is also needed in real-world settings to directly compare different label formats to guarantee more robust evidence of their effectiveness. We have to consider that, on an organizational level, the constraints to implementation are cost and time, particularly for SMEs [8], while the perceived benefits (such as increased competitiveness, benefits for consumers) might not be high enough. There is a need for international harmonization standards on eco-labeling calculations [8]. This issue is still unresolved.

## Figures and Tables

**Figure 1 foods-12-02941-f001:**
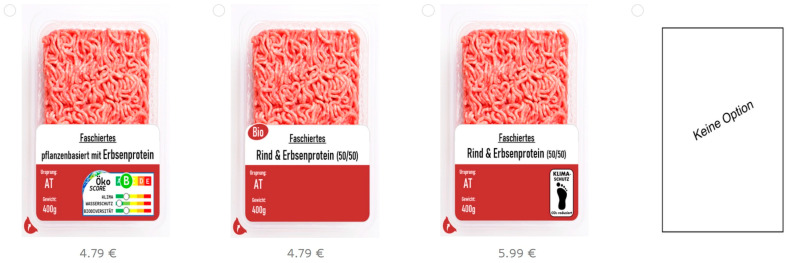
Choice set example with the no-choice option.

**Figure 2 foods-12-02941-f002:**
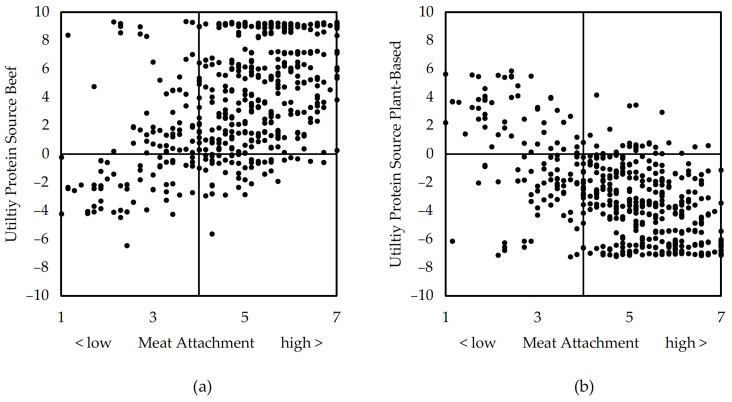
Relationship between MA and the utilities of (**a**) beef and of (**b**) plant-based alternatives.

**Table 1 foods-12-02941-t001:** Sociodemographic characteristics of the full sample (n = 536) and Austria.

		N	Sample%	Austria% ^a^
Consumption group	Meat eaters	510	95.9	94.0
	Vegetarian or vegan	22	4.1	6.0
Gender	Female	275	51.8	51.2
	Male	256	47.2	48.8
Age	18–25	22	4.1	10.6
	26–35	91	17.1	16.4
	36–45	88	16.5	16.0
	46–55	89	16.7	17.6
	56–65	104	19.5	17.2
	>65	139	26.1	22.0
Residence	Rather urban	212	40.2	53.7
	Rather rural	315	59.8	46.3
Highest Education	Mandatory school	112	21.0	21.4
	Apprenticeship, VET school (BMS)	259	48.6	47.1
	High school, college	94	17.6	15.7
	University, academy	68	12.8	15.8
Household Income	Up to 2000 €	155	35.7	30.0
	2001–4000 €	167	38.5	40.0
	More than 4000 €	112	25.8	30.0

^a^ Source: Statistics Austria [71,72,73,74,75].

**Table 2 foods-12-02941-t002:** Attributes and attribute levels tested in the choice experiment.

Attribute	Level	Description
Eco-label	Planet Score B Planet Score D	Fictional comprehensive multi-level label in the style of the Planet Score including the subcategories climate, water protection, and biodiversity
	Carbon Trust—CO_2_ Reduced No label	Fictional binary label “Climate Protection” in the style of the Carbon Trust label
Origin	Austria EU	Geographical origin
Protein type	Beef Beef and pea protein (50/50) Plant-based (peas)	Protein source of minced meat
Production method	Organic Conventional	Most prominent production methods in Austria
Price	3.59 € 4.79 € 5.99 €	Price per 400 g; based on store checks in four major supermarket retailers in Austria

**Table 3 foods-12-02941-t003:** Results from the DCE (n = 490).

Attribute	Variable	Partworth Utility *u_i_* ^a^	SD	95% Confidence Interval ^b^		Mean Relative Importance
				Lower	Upper	
Eco-label	Planet Score B	1.15	0.81	1.07	1.22	17.74%
	Planet Score D	−0.37	0.85	−0.44	−0.29	
	Climate Protection—CO_2_ Reduced	0.22	0.62	0.17	0.28	
	No label	−1.00	0.66	−1.06	−0.95	
Origin	Austria	0.62	0.56	0.58	0.67	9.02%
	EU	−0.62	0.56	−0.67	−0.58	
Protein type	Beef	3.35	3.89	2.99	3.60	44.17%
	Beef and pea protein (50/50)	−0.47	1.53	−0.60	−0.33	
	Plant-based (peas)	−2.88	3.18	−3.16	−2.60	
Production method	Organic	−0.23	0.36	−0.26	−0.20	4.35%
	Conventional	0.23	0.36	0.20	0.26	
Price	β_price_	−1.57		−1.56	−1.57	24.73%
No choice		0.93	2.83	0.67	1.17	

^a^ All partworth utilities *p* ≤ 0.001; ^b^ confidence interval calculated by means of 5000 bootstrap draws.

**Table 4 foods-12-02941-t004:** The WTP in € per minced meat attribute *i*.

Attribute *i*	WTP in €
Planet Score B	+0.73
Planet Score D	−0.23
Climate Protection—Reducing CO_2_	+0.14
No label	−0.64
Origin EU	−0.40
Origin AT	+0.40
Protein source beef	+2.14
Beef and plant-based 50:50	−0.30
Plant-based	−1.84
Conventional	−0.15
Organic	+0.15

**Table 5 foods-12-02941-t005:** Distribution of utilities of protein sources (in %).

	*u_i_* ≤ −2	−2 < *u_i_* ≤ −1	−1 < *u_i_* ≤ 0	0 < *u_i_* ≤ 1	1 < *u_i_* ≤ 2	*u_i_* ≥ 2
Beef	9.6	3.7	8.6	9.0	11.2	58.0
Beef & plant-based 50:50	23.7	19.4	13.1	25.9	12.9	5.1
Plant-based	62.7	11.6	9.6	5.9	1.8	8.4

**Table 6 foods-12-02941-t006:** Explanatory variables and descriptive statistics.

	CA	Mean	Median	SD	Min	Max	N
Subjective Eco-Label Knowledge (ELK)	0.79	5.29	5.5	1.15	1	7	531
Environmental Concern (EC)	0.91	4.94	5.2	1.53	1	7	536
Meat Attachment (MA)	0.84	4.74	5	1.41	1	7	531
Health Consciousness (HC)	0.83	4.97	5	1.24	1	7	533

Scale 1 = “totally disagree”, 7 = “totally agree”.

**Table 7 foods-12-02941-t007:** Correlation matrix and descriptive statistics for ELK, EC, HC, and MA.

	ELK	EC	HC	MA
ELK	1			
EC	0.52 ***	1		
HC	0.39 ***	0.58 ***	1	
MA	−0.13 **	−0.39 ***	−0.43 ***	1
Mean	5.29	4.94	4.97	4.74
SD	1.15	1.53	1.24	1.41
N	531	536	533	531

Scale: 1 = low perceived ELK, EC, HC, MA to 7 = high perceived ELK, EC, HC, MA. Significance: ** *p* < 0.01; *** *p* < 0.001.

**Table 8 foods-12-02941-t008:** Correlation matrix of ELK, EC, HC, and MA and the utilities of the protein source and eco-label attributes.

Utility *u_i_*	ELK	EC	HC	MA
Beef	−0.16 ***	−0.37 ***	−0.29 ***	0.51 ***
Beef & plant based	0.17 ***	0.25 ***	0.13 **	−0.09 *
Plant-based	0.11 *	0.34 ***	0.29 ***	−0.58 ***
Planet Score B	0.05	0.03	−0.04	0.06
Planet Score D	0.13 **	0.17 ***	0.18 ***	−0.15 **
Reducing CO_2_	−0.18 ***	−0.26 ***	−0.18 ***	0.28 ***
No label	−0.07	−0.00	−0.02	−0.14 **

Significance: * *p* < 0.05; ** *p* < 0.01; *** *p* < 0.001; n = 490.

**Table 9 foods-12-02941-t009:** Correlation matrix of ELK, EC, HC, and MA and the importance of attributes.

Importance of Attributes	ELK	EC	HC	MA
Eco-label	0.22 ***	0.34 ***	0.20 ***	−0.18 ***
Origin	0.04	0.09	0.09 *	−0.03
Protein source	−0.06	−0.14 **	−0.02	0.08
Production method	0.10 *	0.17 ***	0.14 **	−0.09
Price	−0.12 **	−0.15 ***	−0.19 ***	0.06

Significance: * *p* < 0.05; ** *p* < 0.01; *** *p* < 0.001.

## Data Availability

The survey data of this study are available on request from the corresponding author.

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
