# Peer review of "Effects of Multi-Level Eco-Labels on the Product Evaluation of Meat and Meat Alternatives—A Discrete Choice Experiment"

_foods, 2023, doi:10.3390/foods12152941_

Round 1
Reviewer 1 Report
The presented manuscript presents an interesting study and the introduction was promising. Nevertheless some problems emerged in the methodological section and in the results section. In this respect due to some analytical choice and errors in the estimation (WTP for example), I did not consider the discussion session, given that results should be re-estimated. Looking at the terminology used, the authors should clearly state if they applyed Conjoint Analysis or Discrete Choice Experiments. The two methodological terms are both used, but given that the authors included a monetary attribute in their design and estimated WTPs, the term Discrete Choice Experiment should be preferred. In this respect please read this article: Jordan J Louviere, Terry N Flynn, Richard T Carson, "Discrete Choice Experiments Are Not Conjoint Analysis", Journal of Choice Modelling, Volume 3, Issue 3,2010,Pages 57-72,ISSN 1755-5345,https://doi.org/10.1016/S1755-5345(13)70014-9.
More in detail the main problems are the following:
Methodology:
-------------
1. It is not clear which experimental design was used: orthogonal, D-efficient, D-optimal etc
2. Is the design balanced?
3. Why the authors decided to present 13 choice sets to each respondent? This is a quite unusual choice and might result in random responses due to "fatigue effects". The authors should have considered blocking the design and presenting 6 choice sets to each block of respondents (2 blocks)
4. The authors used, as they wrote a quite unreal design, where some attributes associations are not very plausible: see Table A.1 profile 8 or 2 for example, with Beef having Planet B score or being labelled as "Reducing CO2". The authors recognise this problem at lines #289-#292, but this lack of realism makes the data collected not trustable.
5. lines #226-230: "“Imagine you are grocery shopping and standing in front of the refrigerated counter. You want to buy minced meat and see available products. We ask you to choose your preferred product in each of multiple rounds. If you normally do not buy minced meat for yourself, imagine choosing for someone else." This is a further aspect that makes the analyses not trustable: only buyers of minced meat should have been interviewed, and not people buying for somebody else, given that the latter kind of purchase does not reflects the respondent preferences.
6. The Formula 2 should be better specified, Formula 3 is wrong, furthermore the authors do not explain in this section how did they treat dummy variables (dummy or effect coded?), and how did they test their hypotheses. The latter aspects emerges only during the analyses in the Results sectino and with few details (which utility was used? average or individual? How did the authors determined utility thresholds (#393)?).
Results:
---------
7. As explained later in these comments, hypotheses could have been tested using interaction variables in the main model. See for example https://doi.org/10.1016/j.ijgfs.2021.100325
8. WTPs: apparently there is a problem in the calculation of WTPs. If you applied formula (3) corrected as for my suggestion, WTPs should be:
PlanetB = -(1.128/-1.488)= 0.76
PlaneD = - (-0.431/-1.488)= -0.3
Organic = -(0.261/-1.488)= -0.175
Even applying the formula as you (wrongly) wrote it, WTPs estimates are wrong.
Please check them all, and re-write all your comments about them.
9. Table 3: It is not clear how you managed to estimate the price coefficient both as categorical and continuos in the same model. Please report it only as continous if you estimate WTPs.
Given that the design choices could not be changed at this stage of the work, my suggestion is to re-think the design, eventually opting for a labelled design (rather than unlabelled) and blocked, recollect the data, and rewrite the paper with a better design (including considering only mince meat buyers). And re-submit the paper.
-----------------------
SPECIFIC COMMENTS
------------------------
[Page 3]:* [56]support
Reviewer note: Space after citation
[Page 5]:* attention check
Reviewer note: Please provide some detail on the “attention check” applied
[Page 5]:* If you normally do not buy minced meat for yourself, imagine choosing for someone else.
Reviewer note: Did you manage to discriminate between those choosing to buy for themself or for somebody else?
[Page 5]:* Based on the product attributes, a reduced study design was calculated by means of the Microsoft Excel add-in XLSTAT (Version 2018.1.1.).
Reviewer note: Which kind of design was used in the research? Which criteria were used?
[Page 7]:* Planet Score grades are not linked to actual product’s environmental impact, which is a slight deviation from objective grading of food products; beef, for instance, would rather not be graded with the Score B, as GHG emissions are in general quite large for producing beef
Reviewer note: Please discuss this choice considering the realism of the choice option presented and more generally of the design used. Can the results of this study be trusted?
[Page 7]:* for
Reviewer note: Maybe “by”?
[Page 7]:* For this study, the utility Vin, is assumed as the linear function [64] of protein type, eco-labels, origin, production method and price. Integrating the selected product attributes, the following utility function of a consumer n for alternative i is approximated according to the additive model in formula 2.
Reviewer note: Formula (2) is not correct. If you considered attribute levels as categorical variables it should be:
Uin = BETAno_choice * NO_CHOICE + BETAlabel1 * LABEL1 + …. + BETAprice * PRICE
Please also specify if the dummies were dummy or effect coded.
[Page 7]:* relative importance of each of the attributes
Reviewer note: The relative importance of the different attributes (not attribute levels) should be calculated as showed in https://doi.org/10.3390/en12132632
[Page 8]:* ??? = β????????? β?????
Reviewer note: Formula (3) is wrong and should be preceded by a minus
[Page 8]:* The validity of the DCE is assumed to be very high.
Reviewer note: This sentence is useless without reporting statistical indexes (McFadden pseudo R2)
[Page 8]:* It means that 80% of all choices were predicted correctly.
Reviewer note: You can only say that the 13th choice was predicted with a 80% probability correctly. You can check the predicted probabilities for every option in each choice set, and then compare them with the real data collected.
[Page 8]:* following
Reviewer note: Following what?
[Page 8]:* Table 3.
Reviewer note: Table 3:
1. for categorical attributes only report (N-1) coefficients (part worth utilities), and specify the reference level
2. Why do you simultaneously treated the price attribute as categorical and continuous? To estimate WTP you should just treat it as continuous
3. If you treated the price attribute as continuous to gather the relative importance you should take the utility as (beta * max(price_level)). Please recalculate or explain how you calculated it
4. Report some statistical indexes of model performance: McFadden pseudo R2, AIC, BIC, LogLikelihood, number of individuals, number of observations, etc
[Page 8]:* 0.818
Reviewer note: Why this is in bold?
[Page 9]:* Willingness to pay (WTP)
Reviewer note: This paragraph should be rewritten after re-estimating the WTPs.
If you applied formula (3) corrected as for my suggestion, WTPs should be:
PlanetB = -(1.128/-1.488)= 0.76
PlaneD = - (-0.431/-1.488)= -0.3
Organic = -(0.261/-1.488)= -0.175
Even applying the formula as you (wrongly) wrote it, WTPs estimates are wrong.
Please check them all, and re-write all you comments about them.
[Page 9]:* The mean, however, is not telling the whole story. Obviously, the majority of consumers prefer beef. The average WTP of -4.18 € simply tells us that, in average, Austrian consumers are not willing to switch away from meat to plant-based alternatives. They are still carnivores.
Reviewer note: To detect clusters of respondents based on their preferences you can apply Latent Class Models to analyze the DCE data.
See for example: https://doi.org/10.1016/S0191-2615(02)00046-2
[Page 10]:* umeat ≥ 2
Reviewer note: How did you choose such threshold of 2 or -2?
[Page 10]:* Hypotheses testing
Reviewer note: 1. Please explain in the methodology section how you tested your hypotheses
2. Why didn’t you test your hypotheses directly in the HB model with interaction variables?
For example “The Planet Score B (vs. Planet Score D) is more preferred from respondents who are more concerned about the environment.” Could be tested creating using the PLANETScoreB dummy plus an interaction variable (PlanetScoreB multiplied by EC): in this way you check the marginal contribution in term of utility of 1 point of EC score on the probability of choosing a product labelled with Planet B Score and you also get the average utility of a product with PlanetScoreB label.
The same approach should be used for the other hypotheses creating the appropriate interactions.
[Page 11]:* utility
Reviewer note: Did you use individuals utility here? Please specify.
[Page 12]:* perceived importance of the attribute
Reviewer note: Given that the perceived importance is derived from utility, in my opinion this analysis is not necessary.
[Page 12]:* 4. Discussion
Reviewer note: This should be: Discussion and Conclusions
[Page 15]:* Figure A1. Choice set example with no-choice option.
Reviewer note: Place this Figure in the Material and Methods section where you introduce you experimental design

Author Response
Dear Reviewer,
thank you very much for the huge effort you made to review our manuscript. We really appreciate you expertise in the field. We tried everything to meet your comments. Please have a look at the attached file.
Once again, thank you very much for your review.
Kind regards
Oliver Meixner & Anna Kolber

Reviewer 2 Report
This manuscript reports a thorough study on effects of multi-level eco-labels in evaluation of meat and meat alternatives (minced), that was done using a discrete choice experiment in an online survey in a representative sample of Austrian adult population. Topic is timely and results are useful for academics and industry in the field. I have only the following minor issues for the Authors to consider for revision.
1. Title. Studying multi-level labels was essential for this manuscript. Consider highlighting that also in the title, for example, by starting the title as "Effects of multi-level eco-labels on --".
2. Abstract. I suggest you add some more key information on the methods to the Abstract: the sample size (number of valid responses), a note about its representativeness (e.g., to the sentence "A Discrete Choice Experiment was conducted in Austria"), and that it was an online survey.
3. Introduction, lines 89-96. Consider moving this paragraph (that tells about your own study) closer to the hypotheses.
4. Lines 201-202. Consider moving the description of the objective of the study from the Materials and Methods section to the end of the Introduction (or before the hypotheses). Note the Instructions for Authors ("Introduction. -- Finally, briefly mention the main aim of the work --"), https://www.mdpi.com/journal/foods/instructions
5. Page 6, first full paragraph. Consider refering here to Appendix Table A3, where you have given the texts of the questionnaire items.
6. Methods, questionnaire in general. Please tell which language was used in the online questionnaire. If it was not English, describe how did you translate the questionnaire items, or cite appropriate sources. Please also consider reporting possible translated texts in Appendix Table A3.
7. Lines 298, 299, and 488. Instead of anonymous expressions such as "proposed by [100]", consider crediting the authors, for example, "proposed by Thurstone [100]".
8. Results in general. Consider how many decimals is optimal for readers to digest the values in Tables and body text. Personally, I find three decimals too many and prefer two decimals in most cases, such as means, correlation coefficients, and Cronbach alpha values.
9. Discussion, line 496-501. It is good that you have discussed not only average effects, but also different consumer segments (target groups). I encourage you to highlight this even more in the Discussion, and perhaps mention it in the Abstract too. Consider discussing your results on consumer segments in the light of previous research on consumers' attitudes to meat alternatives in consumer segments.
10. Your Discussion section now ends to discussion on limitations and future directions. However, I encourage you to form a brief separate Conclusions section (although it is not mandatory for this journal) after the Discussion (or to make it as the last paragraph of the Discussion. In this way, you could better highlight your conclusions to readers.
I'm not a native speaker, but as far as I can evaluate, the language is mostly good and fluent. Standard proofreading could still be needed to make the text even more reader-friendly.
Author Response
Dear Reviewer,
thank you very much for reviewing our manuscript. Attached you will find our reply to your comments.
Kind regards
Oliver Meixner & Anna Kolber

Reviewer 3 Report
Thank you for the opportunity to review the manuscript entitled “Effects of eco-label on the product evaluation of meat and meat alternatives – a discrete choice experiment”, submitted for possible publication to Foods. The purpose of the research is to explore consumers preferences and willingness to pay for a multi-level eco-label by using a discrete choice experiment in Austria. I have some minor concerns, which the authors are kindly asked to address. Please, consider each point-by-point comment.
The abstract is clear and comprehensive. The authors identify the context of the research, highlight the purpose and the methods adopted in the research, and provide interesting insights related to the results. Also, the authors open paths for future research direction and challenges
Keywords are consistent.
I have read with great pleasure the section “Introduction”. It is clear and comprehensive, and the authors have included all relevant aspects to carry out the research. I have only one minor suggestion, which is related to a better structure of the manuscript. First, I would end the section “Introduction” at L. 96, and I would also add some more insights related to the methods applies, the originality of the research and the audience of the study (namely, to whom is the research addressed?). Secondly, I suggest the authors starting, from L. 97, a section entitled “Literature review” or “Theoretical background and hypotheses development”, as to better highlight the importance of the context and the background of the research (the purpose of the section “Theoretical background” is a bit different from the purpose of the “Introduction”).
LL. 128-137 refer to the importance of conducting research in the field of meat and meat alternative. Considering that the authors need to better justify “why” their research focuses on such a food commodity (compared to other commodities, which are rather harmful to the environment), I kindly ask them to include some more statistics and facts, which can immediately help readers understand the reasons for the current research. I agree that the livestock sector accounts for about 14.5% of all GHGs, but could you please include some more data related to resource consumption and environmental impacts? I can suggest the authors the subsequent articles, which are authoritative and are conducted in Europe.
Smetana, S., Ristic, D., Pleissner, D., Tuomisto, H.L., Parniakov, O., Heinz, V. (2023). Meat substitutes: Resource demands and environmental footprints. Resources, Conservation and Recycling, 190, 106831. https://doi.org/10.1016/j.resconrec.2022.106831
Amicarelli, V., Fiore, M., Bux, C. (2021). Hidden flows assessment in the agri-food sector: evidence from the Italian beef system. British Food Journal, 123(13), 384-403. 10.1108/BFJ-05-2021-0547
Ferronato, G., Corrado, S., De Laurentiis, V., Sala, S. (2021). The Italian meat production and consumption system assessed combining material flow analysis and life cycle assessment. Journal of Cleaner Production, 321, 128705.https://doi.org/10.1016/j.jclepro.2021.128705
If the authors are willing to, they could split the section “Theoretical background” into two different subsections, which could be entitled: “Literature review” and “Hypotheses development”, as to better channel the readers’ attention.
The section “Materials and Methods” requires some more information related to the data collection process (L. 204-205). Could you please better specify the period of recruitments and the sampling strategy? I understand that participants have been recruited according to “preselected criteria”, but could you please add more about the sampling strategy, as well as its advantages/disadvantages in terms of comparability and representativeness of the sample? At current, such an information is rather insufficient.
The information related to the participants socio-demographic characteristics should be moved to the beginning of section “Results” (LL. 211-217). Also, could you please justify the assumption: “the sample structure is very close to the structure of the Austrian population” and “we are convinced that results are transferable to the overall Austria population”? A counterfactual, which helps non-Austrian readers understand such assumption, is essential.
LL. 218-230 should be developed further, with specific regard to the “questionnaire structure”.
Overall, I suggest the authors distinguishing in the section “Materials and Methods” at least two different subheadings, namely: “Data collection” and “Data analysis”, as to better channel the readers’ attention and avoid confusion among them.
“Results” and “Discussion” are rather clear and highlight a suitable scientific soundness and a certain utility.
Author Response
Dear Reviewer,
thank you very much for your review. Attached you will find our reply to your comments.
Kind regards
Oliver Meixner & Anna Kolber

Reviewer 4 Report
The presented paper contains vary important results about the consumers willingness to pay and preferences for a multi-level eco label. Thee preferences of the consumers were assessed in regard to meat products, particularly minced meat. In general, my opinion about the manuscript is high, the topis is very timely and relevant, having in mind the requirements for the information of the consumers.
The Abstract of the manuscript concisely describes the main results of the study, and the keywords are properly selected. The Introduction, however, is rather vast. In my opinion, the authors could keep some parts of the Introduction for the discussion, but I will leave this to their decision.
The methodology applied is correct for achieving the aim of the experiment.
The results are clearly described and presented in adequate number of tables. The conclusions, in my opinion, are not well presented. They should get more definition and avoid citing references in the conclusions. In this form, they can not be separated from the discussion. In my opinion, it should be better if they are more clear.
Author Response

(The authors gave the same response as above.)

Round 2
Reviewer 1 Report
I thank the authors for the replies to my previous comments, and the correction they did in the manuscript, like removing not-buyers in the estimates.
The authors addressed some of the comments but did not answer some questions, in order to clarify the soundness of their analyses. Furthermore, my concern about the study design, and in particular the unrealism of the proposed design is unchanged, because the design cannot be modified at this stage of the study --> ("The authors used, as they wrote a quite unreal design, where some attributes associations are not very plausible: see Table A.1 profile 8 or 2 for example, with Beef having Planet B score or being labelled as "Reducing CO2". The authors recognise this problem at lines #289-#292, but this lack of realism makes the data collected not trustable.").
It is like asking somebody to choose between an electric or diesel car, telling that the diesel car has a lower environmental impact. As you might understand, this is implausible, and undermines the full study.
They authors should clarify the following aspects:
---------------------------------------------------
1. Table 23: McFadden Pseudo R2, AIC and BIC were not reported. As it can be seen in this paper (Table 2) https://doi.org/10.1016/j.jhealeco.2008.11.003 , they can be computed also for HB models
2. #346: Formula 2 is still wrong: betas are the estimated coefficients. See here for example (first formula) https://www.sciencedirect.com/science/article/pii/S109830151304391X, or write it as follows: U = Bnc * NoChoice + Sum Bi * Xi + Bcost * COST * e where Sum is the summatory symbol, B is beta, Xi is a vector of the attribute levels
3. #346: the authors did not specify if the attribute levels for the categorical variables were dummy or effect coded. This has important implications on the estimation of WTPs, as explained in https://doi.org/10.1111/ajae.12311 (pages 1776~1778), therefore if the variables were effect coded also formula 3 is wrong, and WTPs should be re-estimated.
4. Table 3: avoid reporting betas for the reference levels attributes in table 3 if you dummy coded your variables. If your attribute levels were dummy coded, these should be zero. I mean No label, EU, Plant-based, Conventional. All coefficient should in fact be interpreted relatively to the omitted level. If you effect coded your variables specify the reference level as done here: http://dx.doi.org/10.1016/j.jval.2016.04.004
5. Table 4: WTP should be interpreted relatively to the omitted level (or reference level) in the estimation. Therefore the reference levels should have a WTP = 0 in the table if your variable were dummy coded. For example it is possible to affirm that respondents are willing to pay € 0.73 more for a product that presents a Planet-B label with respect to a product without label and € 0.96 [WTP = 0.73-(-0.23)] more for a product with Plane-B score with respect to Planet-D score. The authors should anyway clarify if the attribute levels were estimated using dummy or effect coding, because this has consequences on the correct way of estimating the WTPs.
4. #634: change sets with blocks